# A Study on Perceptions towards Organic and Local Production, and Individuals' Socio-Demographic and Geographical Affiliation Influencing Fruit and Vegetable Purchasing Preferences of EU Households

**Alice Varaldo [1], Danielle Borra [1], Emanuela Vassallo [2], Fabrizio Massimelli [2], Stefano Massaglia [1,\*] and Valentina Maria Merlino [1]**

[1] Department of Agricultural, Forestry and Food Sciences, University of Turin, Largo Braccini 2, 10095 Grugliasco, Italy; alice.varaldo@unito.it (A.V.); danielle.borra@unito.it (D.B.); valentina.merlino@unito.it (V.M.M.)

[2] Independent Researchers, Via Vivaro 21/B, 12051 Alba, Italy; e.vassallo@studiovassallo.com (E.V.); f.massimelli@libero.it (F.M.)

\* Correspondence: stefano.massaglia@unito.it

**Abstract:** This study investigates the preferences and the consumption models in the three most relevant F&V EU markets (France, Germany and Italy) in the function of individual attitudes towards local and organic production models. A structured questionnaire was submitted to a sample of 3000 consumers interviewed from December 2021 to January 2022 in the three selected geographical areas. Data were analyzed with the Principal Component Analysis and using the k-means cluster approach. Four main components (European is Better, Organic is Local, Quality is Origin-Related, and Clothes Do Not Make the Product) were defined and used to identify four different consumer clusters (organic and local-sensitive, origin and quality assessment, credence and intrinsic attributes, Global quality evaluation) across the entire sample of consumers. The main findings explain the importance of the origin of the F&V, which, however, is evaluated differently depending on the organic certification, the guarantees made by a brand/logo, and the evaluation of product quality cues. Furthermore, awareness of the logos significantly affects cluster composition and consumption orientation. Finally, the MLR model was applied, highlighting how certain socio-demographic variables, as well as the specific country, explain group membership very well.

**Keywords:** local production; organic; consumer; cluster analysis; Principal Component Analysis; fruit and vegetable

## 1. Introduction

The consumption of locally and organic-managed agro-food is steadily increasing worldwide. The rising attention to such products has found success not only in consumer purchasing but also in food policy issues and research studies [1]. The development concerning the purchase of local and organic fruit and vegetables (F&V) is due to greater attention paid by consumers to their food regime but also thanks to increasing information and awareness campaigns towards a healthier, more balanced diet [2,3]. In addition, organic and local labels open new market opportunities to farmers and suppliers, allowing them higher income for the products grown.

Commonly organic F&V are those grown in agreement with rules that exclude uses of synthetic inputs. Local fruits and vegetables, instead, are described by many definitions. These definitions depend on consumer perception, cultural heritage, and

existing legislation/regulations but always refer to a restricted growing area located near the selling area and connected to environmentally friendly topics [4,5].

These types of accreditations are very common in F&V markets, especially in a context where the consumer has a wide choice of products and is well-informed on the subject [6]. Both local and organic food models, from the point of view of the supply chain and buyers, are considered commercially relevant for their positive effects on the ecology and health, additional economic value, and high-quality assignment. Several studies, in fact, report how consumers are willing to pay a superior price for locally-grown and organically-handled fresh products [7,8].

The demand for local and organic food has always been closely linked to consumers; in fact, it generally tends to associate the two concepts. On the other hand, there is still a divergence between those who prefer to buy local products and those who would rather opt for organic goods. Recent research outlined how buyers have uneven perceptions of agro-food purchasing preferences for products containing the words "organic" and "local" [9]. It remains unclear whether the preferences are complementary or in competition. Thus, it is uncertain whether differences in consumer behavior are related to different eating habits, gender, age and geographical affiliation. In this latter case, the consumer's origin (culture, culinary habits, product availability and awareness) usually defines consumer preference and perceptions of local and organic F&V.

Based on these perspectives, the purpose of this paper was to provide a clear insight into the choices of Italians, the French, and Germans that guide the purchasing of F&V in relation to the importance individuals place on local and organic food production. In other words, the research focused on examining consumer choice inclinations as a function of preferences and awareness of product quality and origin, as well as considering socio-demographic (SD) composition and country of origin detected by an anonymous questionnaire. This investigation aims to understand what, based on buyers' preferences (towards local and organic production) and individual characteristics, mainly affects consumers' purchasing choices and preferences. This purpose is derived from the first research hypothesis (H1), which suggests that there is a relationship between the perception of local and organic food and the customers' buying orientation towards fruit and vegetable quality evaluation, as well as choice habits. To achieve our goal, several aspects, such as the importance of organic and local food, food attributes and packaging preferences, and awareness of different logos (Protected Designation of Origin—PDO, Protected Geographical Indication—PGI and Organic certification) were explored and correlated to the opinion about the European and national F&V products in the three selected countries [10–12]. Moreover, it has also been hypothesized (H2) that the socio-demographic characteristics of consumers, as well as their country of origin, are key factors for the definition of F&V purchasing patterns. With these goals, more precise answers were sought regarding the purchasing preferences of European consumers (Italy, France and Germany) since other studies have shown that there is a discrepancy in purchasing choices based on the age of the consumer, from which we can determine defined groups of buyers based on their SD characteristics [13,14]. The reasons that led to the formulation of these questions are also driven by the fact that the high-quality appellation often lies in the products' origin, which must be the country where the consumers live [15,16].

To reach these objectives, a Principal Component Analysis (PCA) and the Cluster Analysis were performed to identify different consumption patterns and individual clusters. Afterward, the clusters were defined by SD characteristics, country of origin, F&V choice tendencies, awareness, and buying frequency. Finally, an analysis of probability was carried out to predict the cluster membership of consumers' F&V preferences.

## 2. Materials and Methods

### 2.1. Data Collection

Questionnaire data were collected using an online consumer survey conducted in Italy, France, and Germany simultaneously from December 2021 to January 2022 to explore how the importance given to local and/or organic food products affects the European F&V quality perception in the three target countries.

The validated version of this questionnaire was submitted online by a company that specialized in consultancy business, data collection, and analysis of customer experience. The chosen data collection methodology was Computer Assisted Web Interviewing (CAWI).

The survey was conducted in accordance with the ethical standards laid out in the Declaration of Helsinki and was obtained by collecting responses voluntarily from informed individuals over 17 years old.

The questionnaire was dedicated to fruit and vegetable purchasers: in fact, an initial question was designed to ask whether the respondent was the purchasing manager of the household since the questions in the survey focused on choice preferences and purchasing habits. Non-purchasers were discarded from the survey (they could not go ahead with the compilation).

### 2.2. Questionnaire Structure

The questionnaire was anonymous; it excluded sensitive data and was developed in the official language of the country. Furthermore, the survey aimed to profile the total sample in function of their revealed preferences, awareness of product quality and origin, socio-demographic (SD) composition, and individual country affiliation. After the SD section of the questionnaire [16], in Section 2, the respondents were asked about the importance given to specific characteristics that are connected to organic and local food production [17] (food attributes, packaging preferences and awareness of product origin logos) in order to analyze the individual perceptions of organic and local production. The batteries of items selected to develop Section 2 were taken from the literature [18–21]. Given the importance of frequency of purchase in defining consumer preferences [13], the third section of the questionnaire included questions on frequency of F&V purchase and the origin of the purchased products, together with a question on respondents' perceptions of European fruit and vegetables [8,19,21].

The questionnaire content was firstly examined through a preliminary pilot survey to test the internal consistency and reliability of the developed scales using Cronbach's formula considering $\alpha$ values higher than 0.80 as acceptable. The reproducibility was tested using Pearson's correlation analysis. This preliminary analysis allowed us to exclude some items related to the quality perception scale of European F&V (from 10 to 8 descriptors), the attributes considered during F&V purchasing (from 7 to 4), and the packaging characteristics of products (from 7 to 5).

The final version of the questionnaire included four main sections described in Table 1.

**Table 1.** Questionnaire framework. The variable codification employed for the analysis is also reported.

| Questionnaire Section | Variable category | Variables/Question | Codification |
|---|---|---|---|
| Section 1 | Socio-demographic questions | Country | 1 = Italy; 2 = France; 3 = Germany |
| | | Age | 1 = 18–25; 2 = 26–35; 3 = 36–45; 4 = 46–55; 5 = 56–65; 6 =>65 |
| | | Gender | 0 = male; 1 = female |
| | | Family size | 1 = 1 component; 2 = 2 components; 3 = 3 components; 4 = 4 components; 5 => 4 components |
| | | Economic situation | 0 = very difficult; 1 = difficult; 2 = satisfactory; 3 = very satisfactory |
| Section 2 | Importance of organic and local food | • When buying food products, how much importance do you give to organic products? • When buying food products, how much importance do you give to local products? | 5-points Likert scale (1 = not important; 5 = very important) |
| | Food attributes preferences | When buying food products, how much importance do you give to the following attributes? *(packaging; nutritional value; origin of raw materials; production area)* | 5-points Likert scale (1 = not important; 5 = very important) |
| | Food packaging attributes preferences | When buying food products, how much importance do you give to the following packaging attributes? *(images/colors; brands/logos guaranteeing origin and quality; brands/logos guaranteeing a sustainable product)* | 5-points Likert scale (1 = not important; 5 = very important) |
| | Awareness | We will now show you some logos; for each one, please indicate if you are familiar with it (The pictures of PDO, Organic and PGI logos were shown). | 1 = Yes, I know it; 2 = Yes, I know it, but I do not recognize the meaning; 3 = No, I do not know it |
| Section 3 | Frequency of fruit purchase | With reference only to products from the European Union, how often do you buy fruit? | 0 = never; 1 = on special occasions; 2 = a few times a year; 3 = a few times a month; 4 = 1–3 times a week; 5 = 3–5 times a week; 6 = more than 5 times a week |
| | Frequency of vegetable purchase | With reference only to products from the European Union, how often do you buy vegetables? | 0 = never; 1 = on special occasions; 2 = a few times a year; 3 = a few times a month; 4 = 1–3 times a week; 5 = 3–5 times a week; 6 = more than 5 times a week |
| | Opinion about European and national F&V products (How much you | Have a superior quality image; have a high level of safety, thanks to inspections and traceability; | 5-points Likert scale (1 = not in agreement; 5 = totally in agreement) |

| | | |
|---|---|---|
| agree with the following statements? Compared to products from other countries, European and national products…) | taste better; have great nutritional value; are healthy foods; the organic products are better; cost a bit more but I am willing to pay; are more sustainable. | |
| Preference of fruit origin during purchasing decision | Thinking about your fruit purchasing choices; which countries does the fruit you buy come from? (respondents can indicate up to a maximum of three countries) | The respondent indicated the country of fruit origin for each column related to the first, second and third choices. |
| Preference of vegetable origin during purchasing decision | Thinking about your vegetable purchasing choices, which countries does the fruit you buy come from? (respondents can indicate up to a maximum of three countries) | The respondent indicated the country of vegetable origin for each column related to the first, second and third choices. |

### 2.3. Statistical Analysis

All data were recorded in Microsoft® Excel 2010 (Microsoft Corporation Redmond, Washington, DC, USA), and all statistical analyses were performed using SPSS 28.0 for Windows (SPSS Inc., Chicago, IL, USA). Firstly, a descriptive analysis of the sample distributed in the three countries under analysis was made by evaluating the SD aspects by simple tabulation and Pearson's chi-square test ($\chi^2$) to test for dissimilarities among countries ($p < 0.05$).

Consequentially, considering the whole sample, the Principal Component Analysis was performed to obtain different individuals consumption patterns based on the answers about (1) the importance of organic and local food; (2) food attribute preferences; (3) food packaging attribute preferences and (4) the opinion about European and national F&V products. Loading values greater than 0.35 were considered for the factor definition. Cronbach's alpha and Pearson correlation tests were performed for each component to check factor reliability and internal consistency. The correlation test was employed in the case of components explained by less than 2 items. To cluster consumers according to their preferences and perception patterns, the loadings of PCA were employed as variables in the cluster analysis using the k-means technique based on the hierarchical cluster analysis procedure employing the Euclidean distances [22]. Firstly, outlier detection was performed using the box plot visualization tool; then, after cleaning the dataset, we performed a hierarchical CA to estimate the appropriate number of segments (k). This resulted in a scree plot, the node of which indicated that the ideal number of clusters would be 4. The 4-cluster segmentation was judged as good in terms of cohesion and silhouette separation by employing the two-step method using both BIC and AIC indexes. Therefore, the K-means clustering method was applied for cluster definitions. This method assigns rows, in our case represented by consumers, with similar preferences and consumption patterns towards local and organic production and F&V characteristics within the same cluster [23]. The ANOVA and post-hoc Tukey multiple comparison tests were used to verify whether the clusters significantly differ from each other. Each cluster was explored and compared in terms of SD variables, frequency of fruit and vegetable purchase and awareness. Chi-square tests were applied to test the independence between the variables in the different consumer groups. Finally, the association between these letter variables and the cluster membership was assessed using the Multinomial Logit Regression

(MLR) [24,25]. The first model was used, including all the independent variables, following the procedural scheme proposed in Saba et al. (2019) [26]; then, the final MNL models were considered including only the statistically significant ($p \leq 0.05$, Wald chi-square test for model effects) independent variables [27,28]. The $\beta$ value was considered to explain the probability for each predicting variable to belong to a specific cluster in comparison to cluster 4, which we consider to be the reference group. The variables' codification in the MLR was reported in Table 1.

## 3. Results

A total of 3000 individuals were interviewed and were equally distributed in Italy, Germany, and France. The composition of the total sample and of the respondents involved in the three countries analyzed are reported in Table 2.

The total sample ($n$ = 3000) was made up of a balanced percentage of women and men, with many subjects aged between 36 and 55 years of age, equally distributed in families of different sizes and with a satisfactory economic situation. The three target countries, however, differed in terms of SD variables, excluding gender. Individuals belonging to medium-large households, of a medium-high age and with a medium-low economic situation defined the Italian sample. France was represented by purchasers in large families with a medium age and a medium-high economic situation. Finally, the German sample was the oldest group, mostly represented by small families with a similar economic situation if compared to the other countries.

**Table 2.** Sample composition.

| Socio-Demographic Variables | | Country (% within the Country) | | | Total | $\chi^2$ |
|---|---|---|---|---|---|---|
| | | Italy | France | Germany | | |
| Family size (*n*. of components) | 1 | 6.4% | 16.5% | 26.3% | 16.4% | 21.87 * |
| | 2 | 18.9% | 20.9% | 29.8% | 23.2% | |
| | 3 | 34.0% | 26.4% | 24.5% | 28.3% | |
| | 4 | 31.5% | 25.2% | 15.0% | 23.9% | |
| | More than 4 | 9.2% | 11.0% | 4.4% | 32.1% | |
| Gender | Male | 50.6% | 49.8% | 49.3% | 49.9% | 0.344 |
| | Female | 49.4% | 50.2% | 50.7% | 50.1% | |
| Age | 18–25 | 39.9% | 0.3% | 7.4% | 12.4% | 1307.394 *** |
| | 26–35 | 0.0% | 26.6% | 0.0% | 11.2% | |
| | 36–45 | 54.4% | 40.1% | 38.9% | 43.2% | |
| | 45–55 | 5.7% | 33.0% | 34.9% | 26.9% | |
| | >55 | 0.0% | 0.0% | 18.8% | 6.3% | |
| Economic situation | Very difficult | 8.3% | 3.1% | 6.7% | 6.0% | 63.421 *** |
| | Difficult | 33.3% | 31.4% | 31.4% | 32.0% | |
| | Satisfactory | 50.9% | 53.2% | 45.5% | 49.9% | |
| | Very satisfactory | 7.5% | 12.3% | 16.4% | 12.1% | |

Level of significance: *** $p$-value < 0.001; * $p$-value < 0.05.

### 3.1. Consumer Orientation and Preferences towards F&V

The PCA of the consumer ratings for selected variables showed that 65.92% of the variance could be explained by the first four principal components explained in Table 3. For each component, Cronbach's alpha is reported, showing a coefficient always higher than 0.6 [29], indicating appropriate internal consistency. In the case of the PC 4, the Pearson correlation coefficient between the items is reported.

The first component (35.17% of the explained variance), called "European is Better" captures the relation between the European and national origin of fruits and vegetables and the high-quality perception of all the products' characteristics.

More specifically, PC1 expresses a consumption style based on product quality aspects related to intrinsic, extrinsic and sustainability attributes. However, in this component, the significant effect of the importance given to the organic and local production did not emerge.

**Table 3.** Varimax rotated Principal Component Analysis. Each component was named in accordance with the consumption models described by the significance of values.

| Variables | Components | | | |
|---|---|---|---|---|
| | **European Is Better** | **Organic Is Local** | **Quality Is Origin-Related** | **Clothes Do Not Make the Product** |
| Opinion about European and national F&V products (How much you agree with the following statements? Compared to products from other countries, European and national products… | | | | |
| Have a superior quality image | 0.744 | | | |
| Have a high level of safety, thanks to controls and traceability | 0.751 | | | |
| Taste better | 0.747 | | | |
| Have great nutritional values | 0.736 | | | |
| Are healthy foods | 0.769 | | | |
| The organic products are better | 0.584 | | | |
| Cost a bit more but I am willing to pay | 0.628 | | | |
| Are more sustainable | 0.722 | | | |
| Importance of organic and local food | | | | |
| When buying food products, how much importance do you give to organic products? | | 0.799 | | |
| When buying food products, how much importance do you give to local products? | | 0.671 | | |
| Food packaging attributes preferences | | | | |
| Images/colors | | | | −0.419 |
| Brands/logos guaranteeing origin and quality | | 0.601 | | |
| Brands/logos guaranteeing a sustainable product) | | 0.740 | | |
| Food attributes preferences | | | | |
| Origin of the raw materials | | | 0.719 | |
| Nutritional value | | | 0.654 | |
| Production area | | | −0.910 | |
| Packaging | | | | −0.930 |
| *Cronbach's Alpha* | *0.839* | *0.676* | *0.777* | |
| *Pearson correlation* | | | | *r = 0.830* |

Kaiser–Meye–Olkin index = 0.885. Bartlett's sphericity test: Chi-square = 16,549.681; *p*-value = 0.000.

For PC2 (12.53% of the total variance), the "Organic is Local" component links the importance given by the consumer to the organic and local production, simultaneously explaining a connection between the two concepts considering the organic F&V products. In addition, higher quality was perceived in the organic product evaluation, and the quality of products is assessed in this model by means of brands that guarantee their origin, sustainability, and truthfulness. The "Quality is Origin-Related" is the third component (9.53% of the total variance) and describes a choice model purely related to the positive evaluation of the higher nutritional quality of European F&V products, which can be assessed by the area of origin of raw materials.

Finally, the PC4 (Clothes Do Not Make the Product), which accounts for 8.69% of the total variance, on the other hand, can be ascribed to a model of choice characterized by skepticism towards product image and packaging, setting up a pattern of product quality evaluation based on non-external characteristics.

Table 4 describes how the four principal components affect the consumers' cluster definitions. A total of four homogeneous groups were defined and named according to their F&V preferences/perception-related profile, which were derived according to the final cluster centers shown in Table 4. In other words, the indices included in the table explain how (positively or negatively) each principal component influences the cluster definition. For example, in the case of the "Organic and Local Sensitive" cluster, it was positively defined by the patterns "European is Better" and "Organic is Local" components but was not influenced by the other two PCs. Therefore, this cluster was named according to the major influence of the origin together with organic certification in the consumption pattern of its members. This procedure was applied to name the four clusters, in size order: Global quality evaluation, organic and local-sensitive, credence and intrinsic attributes sensitive, and origin and quality assessment. The ANOVA results highlight several significant differences across the four clusters in terms of individual, local and organic perception, preferences, and F&V choice orientations. The biggest cluster, which represents 41% of the total sample, was called "*Global quality evaluation*" and grouped all the individuals oriented to the F&V choice influenced by the overall assessment of quality, paying attention to product origin, intrinsic characteristics, sustainability and certification, but not to the external product features such as the images/colors and packaging.

**Table 4.** Cluster definition.

| | Clusters | | | | |
|---|---|---|---|---|---|
| **Principal Component** | **Organic and Local-Sensitive** | **Origin and Quality Assessment** | **Credence and Intrinsic Attributes Sensitive** | **Global Quality Evaluation** | **F** |
| Cluster size | 29% | 10% | 20% | 41% | |
| European is better | 0.001 [a] | −0.048 [b] | −0.145 [c] | 0.083 [d] | 296.447 *** |
| Organic is local | 0.094 [a] | −0.027 [b] | −1.290 [c] | 0.573 [d] | 324.307 *** |
| Quality is origin-related | −1.320 [a] | 1.163 [b] | 0.332 [c] | 0.473 [c] | 19.091 *** |
| Clothes do not make the product | −0.144 [a] | −2.092 [b] | 0.481 [c] | 0.401 [c] | 768.627 *** |

[a], [b], [c], [d] The mean loading within a row with the same letters is statistically different $p < 0.001$ *** ($\alpha = 0.05$, Tukey's post-hoc test).

The "Organic and Local-sensitive" group (which represents 29% of the total sample) was defined by the consumption model based on the evaluation of quality aspects of F&V, relating them with organic and origin production certification. The third cluster (20% of the sample), named credence and intrinsic attributes sensitive, was composed of consumers who evaluate the origin of the product during the purchasing process considering the raw material origin as an indicator of higher nutritional value of the product. Simultaneously, this consumer group did not evaluate the image and packaging features as important attributes for F&V choice, emphasizing the importance of aspects related to the intrinsic quality and certified origin of products but not to their outward appearance. Finally, the little cluster origin and quality assessment (10% of the total sample) was characterized by individuals who considered the raw material origin the only important attribute that they linked to the higher F&V quality. In this case, the production area was not important for the cluster definition.

### 3.2. Cluster Composition and Membership

With regard to the socio-demographic characteristics, all the considered variables emerged as significantly different among the clusters. The cluster compositions, in terms of all the considered SD, awareness, F&V purchasing frequency, and habit variables, are reported in Table 5.

**Table 5.** Cluster composition.

| Socio-Demographic Variables | | Organic and Local-Sensitive | Origin and Quality Assessment | Credence and Intrinsic Attributes | Global Quality Evaluation | $\chi^2$ |
|---|---|---|---|---|---|---|
| **Socio-Demographic Composition (%)** | | | | | | |
| Economic situation | Very difficult | 4.8% | 12.7% | 5.8% | 4.9% | 19,357 * |
| | Difficult | 31.9% | 46.9% | 34.3% | 29.6% | |
| | Satisfactory | 50.6% | 32.5% | 50.3% | 52.3% | |
| | Very satisfactory | 12.8% | 7.9% | 9.5% | 13.2% | |
| Country | Italy | 19.1% | 30.8% | 44.8% | 30.2% | 175,812 *** |
| | France | 28.2% | 26.8% | 35.1% | 40.2% | |
| | Germany | 52.6% | 42.5% | 20.1% | 29.6% | |
| Age | 18–25 | 12.1% | 11.4% | 16.0% | 10.8% | 70,001 *** |
| | 26–35 | 12.1% | 7.9% | 11.2% | 14.1% | |
| | 36–45 | 40.2% | 42.7% | 49.0% | 40.8% | |
| | 45–55 | 29.5% | 29.8% | 22.9% | 25.9% | |
| | >55 | 6.1% | 8.3% | 0.9% | 8.5% | |
| Gender | Male | 53.5% | 46.0% | 54.8% | 47.9% | 16,612 *** |
| | Female | 46.5% | 54.0% | 45.2% | 52.1% | |
| Family size (*n*. of components) | 1 | 20.3% | 16.3% | 14.7% | 16.1% | 21,487 * |
| | 2 | 24.8% | 24.1% | 23.8% | 20.8% | |
| | 3 | 29.2% | 26.5% | 30.1% | 28.2% | |
| | 4 | 18.0% | 25.6% | 22.8% | 26.0% | |
| | >4 | 7.7% | 7.4% | 8.6% | 8.9% | |
| **Awareness** | | | | | | |
| PDO logo awareness | Yes, I know it | 39.9% | 60.5% | 67.7% | 58.9% | 103.409 *** |
| | Yes, I know it, but I do not recognize the meaning | 21.4% | 17.9% | 14.7% | 14.6% | |
| | No, I do not know it | 38.7% | 21.7% | 17.6% | 26.5% | |
| PGI logo awareness | Yes, I know it | 39.2% | 58.9% | 65.3% | 60.0% | 93.486 *** |
| | Yes, I know it, but I do not recognize the meaning | 22.8% | 19.9% | 16.3% | 16.6% | |
| | No, I do not know it | 38.0% | 21.1% | 18.4% | 23.3% | |
| Organic logo awareness | Yes, I know it | 49.9% | 60.8% | 60.4% | 63.2% | |
| | Yes, I know it, but I do not recognize the meaning | 29.2% | 25.2% | 26.0% | 23.6% | 26.676 *** |
| | No, I do not know it | 21.0% | 14.0% | 13.6% | 13.2% | |
| **Purchasing habits** | | | | | | |
| Frequency of vegetables purchase | 1–3 times a week | 44.6% | 49.7% | 43.4% | 42.9% | 38.184 * |
| | 3–5 times a week | 19.1% | 23.2% | 24.4% | 23.2% | |
| | Never | 1.4% | 0.8% | 0.5% | 0.8% | |
| | >5 times a week | 9.1% | 10.0% | 12.8% | 12.5% | |
| | A few times a month | 23.0% | 13.9% | 15.8% | 16.7% | |
| | A few times a year | 2.1% | 2.4% | 2.5% | 3.2% | |
| | On special occasions | 0.7% | 0.1% | 0.5% | 0.6% | |
| Frequency of fruits purchase | 1–3 times a week | 45.8% | 46.3% | 42.5% | 44.9% | 31.738 * |
| | 3–5 times a week | 17.5% | 23.9% | 22.4% | 21.3% | |
| | Never | 0.9% | 0.6% | 1.3% | 0.2% | |
| | >5 times a week | 11.4% | 11.1% | 13.5% | 14.7% | |
| | A few times a month | 21.0% | 15.3% | 16.8% | 15.9% | |
| | A few times a year | 2.7% | 2.3% | 3.2% | 2.1% | |
| | On special occasions | 0.7% | 0.4% | 0.3% | 0.9% | |
| First choice of country of origin during fruit purchase | Argentina | 0.9% | 0.6% | 1.2% | 0.7% | |
| | Australia | 0.5% | 0.4% | 0.4% | 0.8% | 187.013 *** |
| | Belgium | 1.6% | 0.9% | 0.5% | 0.7% | |

| | | | | | |
|---|---|---|---|---|---|
| | Chile | 1.4% | 0.4% | 1.5% | 0.5% | |
| | France | 22.1% | 20.9% | 28.2% | 31.1% | |
| | Germany | 31.9% | 25.1% | 10.5% | 19.9% | |
| | Italy | 16.4% | 30.3% | 37.4% | 26.7% | |
| | No answer | 7.7% | 5.2% | 7.8% | 5.6% | |
| | New Zealand | 1.1% | 0.4% | 1.1% | 1.1% | |
| | Netherland | 1.1% | 2.4% | 0.8% | 1.7% | |
| | Spain | 11.6% | 12.0% | 8.6% | 9.6% | |
| | South Africa | 2.7% | 1.0% | 1.7% | 1.4% | |
| | USA | 0.9% | 0.4% | 0.4% | 0.3% | |
| First choice of country of origin during vegetable purchase | Argentina | 0.5% | 0.5% | 0.5% | 0.9% | |
| | Australia | 0.5% | 0.3% | 0.4% | 0.3% | |
| | Belgium | 0.9% | 0.9% | 0.7% | 0.6% | |
| | Chile | 1.6% | 0.3% | 0.9% | 0.3% | |
| | France | 22.3% | 23.0% | 30.7% | 31.8% | |
| | Germany | 33.9% | 27.6% | 11.9% | 20.8% | |
| | Italy | 17.3% | 31.9% | 38.4% | 27.9% | 185.487 *** |
| | No answer | 8.2% | 4.8% | 7.8% | 5.1% | |
| | New Zealand | 0.2% | 0.4% | 0.1% | 0.7% | |
| | Netherland | 3.6% | 2.5% | 1.3% | 4.2% | |
| | Spain | 10.3% | 6.9% | 6.4% | 6.4% | |
| | South Africa | 0.2% | 0.5% | 0.4% | 0.6% | |
| | USA | 0.5% | 0.3% | 0.4% | 0.3% | |

Significance level: *p*-value < 0.01 *; <0.001 ***.

In order to understand the association between the predictor variables and the cluster fitting, a final MRL model was performed by including those variables that were statistically significant (Wald chi-square test, *p*-value < 0.05) (Table 6) in a preliminary model (data not shown), in which all variables shown in Table 5 were included [26]. The analysis of the probability (β) of each predicting variable being associated with each group was performed considering the Global quality evaluation as the reference cluster, which was omitted from the table of MLR results [30].

**Table 6.** The results of multinomial logistic regression analysis considering the significant socio-demographic awareness and habitudinal variables as a predictor of cluster membership are reported.

| Predictor Variables | Organic and Local-Sensitive | | Origin and Quality Assessment | | Credence and Intrinsic Attributes | |
|---|---|---|---|---|---|---|
| | β | Std. Error | β | Std. Error | β | Std. Error |
| Intercept | **−1.456 *** | 0.314 | −0.185 | 0.229 | **−0.388 *** | 0.155 |
| (Age group = 1] | **1.424 *** | 0.366 | 0.326 | 0.286 | **2.188 *** | 0.520 |
| [Age group = 2] | **1.170 *** | 0.365 | −0.008 | 0.290 | **1.714 *** | 0.521 |
| [Age group = 3] | 1.052 | 0.301 | 0.426 | 0.229 | **2.146 *** | 0.492 |
| [Age group = 4] | 0.988 | 0.305 | **0.514 *** | 0.232 | **1.910 *** | 0.496 |
| [Country = 1] | **−1.435 *** | 0.225 | −0.238 | 0.169 | **0.476 *** | 0.182 |
| [Country = 2] | **1.012 *** | 0.173 | **−0.453 *** | 0.142 | **0.281 *** | 0.161 |
| Awareness | | | | | | |
| [PDO Awareness = 1] | −0.216 | 0.192 | **0.400 *** | 0.161 | **0.701 *** | 0.176 |
| [PDO Awareness = 2] | 0.254 | 0.204 | **0.490 *** | 0.180 | **0.557 *** | 0.200 |
| [PGI Awareness = 1] | **−0.676 *** | 0.200 | −0.098 | 0.169 | −0.018 | 0.182 |
| [ORGANIC Awareness = 1] | **−0.387 *** | 0.176 | −0.131 | 0.152 | −0.222 | 0.162 |
| Frequency of F&V purchase | | | | | | |
| [Fruits-frequency of purchase = 0] | 1.609 | 1.221 | 1.840 | 1.122 | **3.918 *** | 1.233 |

| | | | | | | |
|---|---|---|---|---|---|---|
| [Fruits-frequency of purchase = 2] | 0.735 | 0.806 | 0.695 | 0.714 | **1.837 \*** | 0.874 |

Note: the "Global quality evaluation" cluster was considered as the reference for comparison with the other consumer groups. Model Fit Statistics: Nagelkerke Pseudo $R^2$ = 0.148. Full model $\chi^2$ (df = 96) = 351.401, *p*-value < 0.001. Classification accuracy (90.2%). Significance level: *p*-value < 0.01 \*; <0.05 \*\*; <0.001 \*\*\*.

Considering the socio-demographic features, the probability of belonging to the "organic and local-sensitive" group was significantly higher for consumers with a low/medium age rather than the reference group. In fact, the reference group was composed of mostly older consumers. At the same time, all the age groups, except the older one, seemed to have a greater probability of being associated with the credence and intrinsic attributes cluster. This latter result is in accordance with the age cluster composition.

Furthermore, individuals originating from Italy and France had a low probability of belonging to the organic and local-sensitive cluster but a stronger probability of being associated with the credence and intrinsic attributes cluster with regard to the Global quality evaluation. On the contrary, France was associated negatively with the "origin and quality assessment".

The awareness of the PDO logos appeared positively for origin and quality assessment and credence and intrinsic attributes with regard to the reference cluster. Considering the Odds ratio coefficient, a unit increase in the variable PDO awareness (level 1 = Yes, I know it) (OR = exp β = 1.49) increased the probability of belonging to the origin and quality assessment group by 49% (1-OR). At the same time, for the same variable predictor, the probability of belonging to the cluster credence and intrinsic attributes (OR = exp β = 2.01) increases by over more than 101% as one unit of awareness grows. On the contrary, the consumer who was aware of the PGI logo had a high probability of not belonging to the organic and local-sensitive cluster. By analyzing the odds ratios calculated for this variable, a unit increase in this factor decreased the probability of belonging to the cluster by 96% (1-OR = 1–1.96). Furthermore, an awareness of the organic logo equal to 1 (Yes, I know it) emerged as negatively associated with organic and local-sensitive, with regard to the global quality evaluation.

Considering the fruit purchasing frequency predictor variable, the consumers characterized by a low or non-purchasing rate were positively associated with credence and intrinsic attributes.

## 4. Discussion

These findings show the effectiveness of consumer perception and awareness of organic and local production in measuring the quality evaluation of F&V, as well as the relationship between individual preferences and SD, geographical affiliation, and purchasing habits. Considering the consumption patterns obtained based on the importance of organic and local food, the opinion of European and national F&V products, and the food and packaging attributes, the results indicated different clusters of European consumers.

Firstly, four different consumption patterns emerged from the PCA. The first one, "European is Better", expresses a consumption pattern in which individuals emphasize the origin of food from the country they live in and associate it with a product-vision of high quality, safety, and sustainability [31]. This tendency was also found in a study conducted on consumer preferences for fruit and vegetables from which a group of consumers (called proposed loyalty) [16] emerged; said group is attentive to product quality characteristics together with national origin and certification of the origin of the fruit and vegetables. In this case, the attention and evaluation of the origin of the products, linked to the perception of superior quality, express consumer confidence in proximity production, which is more controlled, safer, and of higher quality. In our research, therefore, a pronounced role of local origin emerged in the "Quality is Origin-Related" component,

which, however, was dissociated from the positive evaluation of information on the production area. This result is in agreement with the research in [15], which indicated a discrepancy between the preference for origin and the indication of area and production method. In fact, in this consumption pattern emerged that the origin of raw materials was the only discriminating factor for quality evaluation. In this case, the attention and evaluation of the origin of the products, linked to the perception of superior quality, express consumer confidence in the proximity production, which is more controlled and safer.

The "Organic is Local" pattern was connected to a choice model based on the evaluation of the local and organic products, which are assessable through a label that guarantees their truthfulness [32]. This result could justify choice fidelity towards a quality and origin label; here, quality aspects become prerequisites, as already found in [33]. In this case, the area of production is important in the definition of the consumption pattern. The description of this principal component is found in Massaglia et al. [16]. Here, the connection between the importance given to local origin and organic certification in the process of choosing fruit and vegetables emerged in a defined consumer group [13]. Finally, the last component "Clothes Do Not Make the Product" described a pattern of F&V choice for which the evaluation of the external components of the product takes second place. This result may stem from the type of product that is often marketed without a brand, label or packaging [16].

On the achievements, it is, therefore, possible to consider the first assumption (H1) accepted since the PCA analysis highlights the presence of four different consumer models that outline different behaviors based on F&V "organic" and "local" appellatives. This confirms the buyers' uneven perceptions towards agro-food purchasing preferences with the "organic" and "local" appellations already found in other studies [9]. This result defines how the connection between organic and local is true both in more limited geographical contexts, such as the one explored in Massaglia et al. [16] but also at the European level. In fact, this local/organic relationship has been extensively studied in several countries and has been found to be decisive in the choice of certain food products, such as fruit and vegetables specifically [4,5,8,21,34,35]. For example, in a study conducted in Northern Europe, the same product opinions emerged that motivated organic and local food consumers [1]. However, the same study highlighted that, although it is the same preference for authentic food that motivates consumers of local and organic foods, differences in terms of individuals' characteristics emerged between the two groups of consumers. The latter result also confirms our findings of diversity between the consumer groups linked specifically to the organic certification and those related only to the local production. For example, the "origin and quality assessment" cluster, explained uniquely by the component quality is origin-related, was mainly composed of people in a difficult financial situation or mature people living mainly in Germany and Italy, two countries with a higher percentage of consumers aware of the meaning of the logos of guaranteed origin. This group, in general, highlights a common search for quality and safety, which can be found even in products with a low added value, but which are still of great quality because they are produced close to home.

Awareness of the geographic indication logos can probably be justified by the higher percentage of mature individuals within this group, also defined as the most probable predicting category of the MRL. These consumers also showed high awareness of the organic certification logo. In this case, the increasing search for healthy and sustainable products by older people, attributes often linked to organic production, can be connected with this latter result, as already found in [14].

Surprisingly, on the contrary, the "organic and local-sensitive" clusters, defined by the "Organic is Local" component, were made up of individuals mainly distributed in the satisfactory economic bracket, predominantly from Germany and France; in general, they showed the lowest awareness of the origin and organic certifications logos in comparison to the other three clusters and displayed important differences in terms of socio-demo-

graphic characteristics and F&V purchasing habits. This result highlights the clear relationship between consumer awareness and the individuals' countries of origin but also identifies a consumer gap and misinformation towards a logo that actually represents a type of production it claims to be important. This result was explained by Kaczorowska et al. [36], who found a discrepancy between consumers' low awareness, positive attitudes, and intention to purchase sustainable products. In addition, other researchers also found that lower-medium economic availability, together with misinformation, defines consumption patterns far removed from the importance of certified quality. Regarding this cluster, we can confirm the results of other authors who in previous research had already found how logo awareness decreases in relation to the individuals' age [37,38]. However, France was also well represented by consumers in the global quality evaluation cluster (the biggest group), composed mainly of young people, with a high level of consideration towards logos of origin and quality and organic certification. The pattern of choice that characterized it, defined by an overall assessment of product quality, both organic and local, could be justified by the origin (France is the second European country in terms of the number of certified food products) [39] and by the average age of the consumers. In fact, it seems that young and middle-aged people have greater sensitivity and curiosity for certified foods with a high added value, perhaps emphasizing the desire to seek out quality niche foods [40,41], often linking them with sustainable attributes [42]. This consumption pattern can also be found in the credence and intrinsic attributes cluster, also represented by many young people in general and Italians specifically. In this case, origin seems to be important in the definition of this cluster. In fact, consumers seem to be more influenced by product quality and origin, which also seems to be defined by a high level of awareness and high purchase frequency of fruit and vegetables, as also revealed by the MLR. This cluster could possibly be associated with a healthy lifestyle, given the focus on origin-defined products with high nutritional value. In addition, when analyzing consumer choices per country, interesting differences emerge regarding the choice of the country of origin of fruit and vegetables during the purchasing process: in fact, while in Italy and France, respectively, 77% and 75% of the consumers interviewed in the individual countries stated that they chose their country of origin as their first choice, only 61% of the Germans stated that they chose fruit produced in Germany as their first option. For vegetables, the percentages were also comparable. This result defines the heterogeneity of the evaluation of the importance of the national origin of fruit and vegetables for European consumers living in different countries, which can be confirmed by the patterns and choice profiles revealed in this research.

Based on the obtained results, H2 can be deemed granted since the consumers' SD affects purchasing habits. In fact, different types of choices emerged according to age, the consumers' income, and their geographical affiliation. These patterns are in line with other studies that highlighted the same tendency of customers to make purchases based on their SD characteristics [13,14].

F&V proximity-sensitive consumers seem more aware of origin and quality logos, as already found in [12]. Goudis and Skuras highlighted that countries such as France and Italy were characterized by a high logo awareness linked with many registered Geographic indications.

## 5. Conclusions

This paper evaluated the factors that might affect consumers' F&V purchasing preferences in a sample of 3000 buyers in Italy, France, and Germany. The main research findings highlighted that, generally, the guarantee of a high-quality product is its origin label, which is assessed differently from organic-certified products. Indeed, what emerged is that origin mostly influences the consumer's choices. Nonetheless, this tendency is not homogeneous in the four clusters discovered during the studies, thanks to the components defined by the PCA. The clusters described the consumers' orientation as well as

purchasing preferences and considered the SD composition, country of origin, customers awareness, and purchasing habits.

As a result of data processing, it was possible to distinguish four purchasing behaviors. The largest percentage of the sample was found to be individuals with a satisfactory economic situation who place their attention on all quality attributes, pay attention to the origin, certifications, and product image. There is also a percentage of consumers who value a quality product only if its origin and organic management are certified. Attributes such as raw material origin certification were found to be a discriminating factor on purchase choices for a group of young consumers with very good awareness of European Union quality logos. However, there is also a segment of consumers, represented by generally older individuals in a difficult economic situation, who identify a high-quality product if it is linked to its origin. In fact, image packaging and raw materials are important food attributes to them.

Although the research allowed to distinguish the characteristics that most affect consumer choices, such as origin, packaging, and image, from those considered less important, such as nutritional and organic values, future studies should focus on the main F&V purchase locations. In this way, it could implement the limits of research that is lacking in surveys on the main distribution channels used by consumers for buying local and organic foods. In addition, the questionnaire used for research purposes could be improved in geographical terms by extending the study to other important countries for the European economy and F&V consumption. This research could provide implications for the policy support of localized food systems and for the future development of this sector from an international market perspective. Indeed, while farmers have been called upon to engage in more direct relationships with end consumers in order to produce and market products on a local basis, on the other hand, the market for fruit and vegetables, especially organic ones, is large and heterogeneous and has to address the needs of differentiated target consumers. In policy terms, the study of consumers' motivations and orientations of choice makes it possible to address recommendations for market orientation.

## 6. Future Research

This research provided interesting results on consumer perceptions of local and organic production and how this influences fruit and vegetable choices by a sample of 3000 European consumers intercepted in three different countries, Italy, Germany and France. The analysis of the answers obtained from the extended sample was conceived in order to define how the socio-demographic characteristics of individuals, together with their geographical affiliation, influence the definition of different F&V styles, profiles, and choice orientations. To this end, a model was developed to combine different statistical methodologies for examining consumer choice inclinations according to preferences, awareness of product quality and origin, as well as considering socio-demographic composition, purchasing habits, and country of origin within the obtained consumption profiles. In this research, in fact, we treated the geographic affiliation of individuals as a variable, along with socio-demographic characteristics, as this dataset was the beginning of a larger collection procedure that now also involves non-European countries. Consequentially, this research will be developed in the future by emphasizing cross-country comparisons in order to highlight the differences in terms of perception, preferences, awareness, and purchasing habits of fruit and vegetables by comparing different countries within and outside Europe.

**Author Contributions:** Conceptualization, S.M., D.B., V.M.M. and A.V.; methodology, V.M.M. and A.V.; software, V.M.M.; validation, V.M.M. and A.V.; formal analysis, AV.M.M. and A.V.; investigation, E.V. and F.M.; resources, E.V. and F.M.; data curation, E.V., S.M., D.B. and F.M.; writing—original draft preparation, V.M.M. and A.V.; writing—review and editing, V.M.M., S.M. and A.V.; project administration, E.V., S.M. and F.M. All authors have read and agreed to the published version of the manuscript.

**Funding:** This research received no external funding.

**Institutional Review Board Statement:** Not applicable.

**Informed Consent Statement:** Informed consent was obtained from all subjects involved in the study.

**Data Availability Statement:** Not applicable.

**Conflicts of Interest:** The authors declare no conflict of interest.

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
