# Peer review of "A Study on Perceptions towards Organic and Local Production, and Individuals’ Socio-Demographic and Geographical Affiliation Influencing Fruit and Vegetable Purchasing Preferences of EU Households"

_horticulturae, doi:10.3390/horticulturae8080670_

Round 1

Reviewer 1 Report

The authors examine the attitudes of French, German and Italian consumers towards Fruits and Vegetables purchasing habits and preferences. Its effect is a comprehensive study, which is an example of reliable scientific work. Their results can help outline strategies on local and organic F&V products. The conclusion contains the main results showing that the aim of the study was achieved.

I find the issue relevant and have not objectively identified major shortcomings.

In my opinion, the article in its current form is suitable for publishing, although due to the obligation of the reviewer I would like to make a few minor comments:

In my opinion, addressing the issues listed below could improve the article.

How did you design the questions (i.e. process to define the claims)? Also, the authors need to provide justification of the study design.

Line 270-274: The authors wrote that “In our research, therefore, a pronounced role of local origin emerged, which, however, in the case of this component, was dissociated from the positive evaluation of information on the production area. This result is in agreement with research [23]  in which there was a discrepancy between the preference for origin and the indication of area and production method..”

The same argument is repeated in the 281-285 - consider the removal of this paragraph.

In the line 348  the word “Always”  should be removed.

“Always Goudis and Skuras highlighted as country like France and Italy were characterized by a high logo awareness linked with many registered many Geographic indications.”

The authors could also indicate the policy implications of this study.

Author Response

Reviewer #1

Rev: The authors examine the attitudes of French, German and Italian consumers towards Fruits and Vegetables purchasing habits and preferences. Its effect is a comprehensive study, which is an example of reliable scientific work. Their results can help outline strategies on local and organic F&V products. The conclusion contains the main results showing that the aim of the study was achieved.

I find the issue relevant and have not objectively identified major shortcomings.

In my opinion, the article in its current form is suitable for publishing, although due to the obligation of the reviewer I would like to make a few minor comments:

Authors: Thank you very much for your comments and revisions. We are grateful for your consideration.

Rev: In my opinion, addressing the issues listed below could improve the article.

How did you design the questions (i.e. process to define the claims)? Also, the authors need to provide justification of the study design.

Authors: thank you for the suggestion. We agree. We revise the materials and methods section including the explanation of the study design and development (see lines 84-92 of the revised manuscript).

Rev: Line 270-274: The authors wrote that “In our research, therefore, a pronounced role of local origin emerged, which, however, in the case of this component, was dissociated from the positive evaluation of information on the production area. This result is in agreement with research [23] in which there was a discrepancy between the preference for origin and the indication of area and production method...”

The same argument is repeated in the 281-285 - consider the removal of this paragraph.

Authors: yes, we removed the paragraph. Thank you!

Rev: In the line 348  the word “Always”  should be removed.

“Always Goudis and Skuras highlighted as country like France and Italy were characterized by a high logo awareness linked with many registered many Geographic indications.”

Authors: yes, thank you.

Rev: The authors could also indicate the policy implications of this study.

Authors: yes, thank you for the suggestion. We implemented some policy implications at the end of the conclusion section (see lines 426-433 of the revised manuscript).

Reviewer 2 Report

Thank you for this interesting paper. Doing this study was a hard piece of work! Unfortunately, however, this paper does not discuss the different countries at all. The paper would gain considerably if the analyses carried out had been done separately for each country and examined in terms of differences. It must be assumed that the respondents in the three countries show clear differences in terms of their preferences and behaviour. By looking at the total sample here, relevant results remain undiscovered. This is also the only reason why I would reject the paper unless it is fundamentally revised. Differentiate your analyses by country and I would support adoption. In doing so, please also note the references I give to the individual chapters.

Good luck!

KEYWORDS:

I suggest to include F+V to the keywords as well as PCA

INTRODUCTION:

The first sentences of the introduction seem to contradict the first sentence of the abstract. Please check again carefully whether you really want to write it this way.

The introduction is comparatively general. Here it would be good if the authors, on the one hand, included a relevant portion of the literature used in the discussion. On the other hand, I recommend posing hypotheses that will be discussed in the discussion.  This would automatically also structure the discussion better.

DATA COLLECTION:

Please restructure this chapter. It contains some paragraphs that clearly fit  better to the statistical analysis than to the data collection.

Please indicate who exactly you interviewed. Citizens? Consumers of fruit and vegetables? People who buy fruit and vegetables? Or people who consume and buy fruit and vegetables? Please explain briefly why you chose the specific group.

Table 1:

Why was a  10 point Likert scale used at one time and a 5 point scale at another? If it was always a 10 point scale, please correct the information with the 5 point scales. Why was a  10point scale used?  This is comparatively untypical. Please explain why you think that the respondents were not overwhelmed by this. As far as I know, it becomes difficult for respondents to answer more than a seven point Likert scale.

Furthermore, it remains unclear why you ask European and domestic products together in your questions and do not differentiate here. Do you think that domestic products are assessed in the same way as European products? I strongly doubt this, as the literature on a wide variety of products repeatedly comes to the conclusion of how much national products are preferred by consumers.

STATISTICAL ANALYSIS:

Was an outlier analysis carried out before the actual cluster analysis? If not, why not? This should definitely be done!

It seems that no discriminant analysis was carried out to check the goodness of the identified clusters. This should definitely be done!

RESULTS:

Table 2:

In order to assess the quality of the sample, it would be good if you could provide data on the distribution of the SD in the respective countries.

Why did you choose a Varimax rotation? Please give reasons. Did you check the items regarding MSA and KMO? Were all items suitable for PCA?

What about Cronbach's Alpha for the PCs? Are the found PCs reliable? Please check and add.

Many items load on two or more PCs, which at first glance speaks against the results found with regard to the discriminatory power of the identified PCs.

The rationale for naming the clusters is very brief. It would help readers who are less familiar with the method if it were explained in more detail how you arrived at the names. Please explicitly include Table 4 for this purpose.

DISCUSSION

The discussion is well written for the present paper (consideration of the total sample). Here, after re-analysis, it would be important to write country-specifically.

The literature used here should also be considered, at least to a large extent, in the introduction.

Author Response

Reviewer #2

Rev: Thank you for this interesting paper. Doing this study was a hard piece of work! Unfortunately, however, this paper does not discuss the different countries at all. The paper would gain considerably if the analyses carried out had been done separately for each country and examined in terms of differences. It must be assumed that the respondents in the three countries show clear differences in terms of their preferences and behavior. By looking at the total sample here, relevant results remain undiscovered. This is also the only reason why I would reject the paper unless it is fundamentally revised. Differentiate your analyses by country and I would support adoption. In doing so, please also note the references I give to the individual chapters.

Good luck!

Authors: Thank you very much for this observation and, generally, thank you for your constructive and useful suggestions. Indeed, we agree that the work could have been done differently in order to better compare the different countries. However, since this data was the beginning of a broader data collection that is now also involving non-European countries, we will certainly try to emphasize a cross-country comparison in the work that will result from the whole dataset. In fact, this research was designed to analyses a unique sample characterized by different components, including geographical affiliation. We conceived it as an analysis of an extended sample of European consumers. We believe that this work has provided interesting results regarding consumers' perception towards local and organic production and how it affects individuals' choices. Furthermore, considering the whole sample, this research has allowed us to define how the socio-demographic characteristics of individuals, together with their geographical affiliation, affect the definition of F&V choice orientations. It is also true that the title of the article could be misleading, as the comparison between countries was not the main purpose of the research. Given that the objective of our work already was "… the research focused on examining consumer choice inclinations as a function of preferences and awareness toward product quality and origin, as well as considering socio-demographic (SD) composition and country of origin detected by an anonymous questionnaire… To reach this objective, the Principal Component Analysis (PCA) and the Cluster Analysis were performed to identify different consumption patterns and individual clusters. Afterwards, the clusters were defined by SD characteristics, country of origin, F&V choices tendency, awareness and buying frequency…" and indeed we treated the geographic affiliation of individuals as a variable, along with socio-demographic characteristics, following the model already proposed by other authors (Ahmad 2010; Moraga and Malek 2020; Perera et al. 2020; Kokthi et al. 2021; Budhathoki and Pandey 2021; Cavite et al. 2022; Merlino et al. 2022), the work was kept as such by changing the title in “A study on perceptions towards organic and local production, individuals' socio-demographic and geographical affiliation influencing the fruit and vegetable purchasing preferences of EU households” to avoid misunderstandings, and eliminating from the text the references to cross-country comparison. We have therefore tried to emphasize in the text the use of the variable 'country of origin' as an incidental factor in defining consumers' preferences and attitudes towards local and organic production. We hope that the re-definition of the work, keeping the objective as such, will convince you of the validity of the research. In fact, we are also convinced that the combination of the selected statistical methodologies could enhance the data in our possession. In particular, the use of PCA and cluster analysis for the total sample provides interesting results. The analysis of the results by country will certainly provide important implications, but should, in our opinion, involve the use of different methodologies such as comparing means and correlations.

References:

  • Ahmad SNBB. 2010. ORGANIC FOOD: A STUDY ON DEMOGRAPHIC CHARACTERISTICS AND FACTORS INFLUENCING PURCHASE INTENTIONS AMONG CONSUMERS IN KLANG VALLEY, MALAYSIA. IJBM. 5(2):p105. https://doi.org/10.5539/ijbm.v5n2p105
  • Budhathoki M, Pandey S. 2021. Intake of Animal-Based Foods and Consumer Behaviour towards Organic Food: The Case of Nepal. Sustainability. 13(22):12795. https://doi.org/10.3390/su132212795
  • Cavite HJ, Mankeb P, Kerdsriserm C, Joedsak A, Direksri N, Suwanmaneepong S. 2022. Do behavioral and socio-demographic factors determine consumers’ purchase intention towards traceable organic rice? Evidence from Thailand. Org Agr. 12(2):243–258. https://doi.org/10.1007/s13165-022-00387-1
  • Kokthi E, Canco I, Topulli E, editors. 2021. Whose salad is organic? An attribute segmentation perspective-evidence from Albania. Economia agro-alimentare / Food Economy.
  • Menezes E, Deliza R, Chan HL, Guinard J-X. 2011. Preferences and attitudes towards açaí-based products among North American consumers. Food Research International. 44(7):1997–2008. https://doi.org/10.1016/j.foodres.2011.02.048
  • Merlino VM, Renna M, Nery J, Muresu A, Ricci A, Maggiolino A, Celano G, De Ruggieri B, Tarantola M. 2022. Are Local Dairy Products Better? Using Principal Component Analysis to Investigate Consumers’ Perception towards Quality, Sustainability, and Market Availability. Animals. 12(11):1421. https://doi.org/10.3390/ani12111421
  • Moraga MJB, Malek S. 2020. Identifying Chilean Dietary Patterns, Key Groups and Sociodemographic Drivers in Aims to Meet The Sustainable Development Goal 12.
  • Perera G, Sprechmann M, Bourel M. 2020. Benefit segmentation of a summer destination in Uruguay: a clustering and classification approach. Journal of Tourism Analysis: Revista de Análisis Turístico. 27(2):185–206. https://doi.org/10.1108/JTA-07-2018-0019

Rev: KEYWORDS:

I suggest to include F+V to the keywords as well as PCA

Authors: thank you! We added the suggested keywords

Rev: INTRODUCTION:

The first sentences of the introduction seem to contradict the first sentence of the abstract. Please check again carefully whether you really want to write it this way.

Authors: We agree. We are sorry for creating confusion. We have removed the first sentence in the abstract while keeping the sentence in the introduction, which is justifiable by reference.

Rev: The introduction is comparatively general. Here it would be good if the authors, on the one hand, included a relevant portion of the literature used in the discussion. On the other hand, I recommend posing hypotheses that will be discussed in the discussion.  This would automatically also structure the discussion better.

Authors: Thank you very much for these suggestions. We have tried to define the research hypotheses and discuss the results according to them in the discussion section, also adding some references. In general, following several changes in the text, the discussion has been revised in several parts. 

Rev: DATA COLLECTION:

Please restructure this chapter. It contains some paragraphs that clearly fit better to the statistical analysis than to the data collection.

Authors: we agree. We revised the chapter following your suggestion and including a new sub-section entitled “questionnaire structure”. Thank you.

Rev:

Please indicate who exactly you interviewed. Citizens? Consumers of fruit and vegetables? People who buy fruit and vegetables? Or people who consume and buy fruit and vegetables? Please explain briefly why you chose the specific group.

Authors: we agree. The questionnaire asked whether the respondent was the purchasing manager of the household, as the questions on choice preferences were focused on the moment of the purchasing decision. Non-purchasers were discarded from the survey (they could not go ahead with the compilation). We clarified this in the text (see lines 118-122 of the revised manuscript).

Rev: Table 1:

Why was a 10 point Likert scale used at one time and a 5 point scale at another? If it was always a 10 point scale, please correct the information with the 5 point scales. Why was a  10point scale used? This is comparatively untypical. Please explain why you think that the respondents were not overwhelmed by this. As far as I know, it becomes difficult for respondents to answer more than a seven point Likert scale.

Authors: thank you very much for the observation. We revise the refuse and the scale indication unifying all in 5-points Likert scale.

Rev: Furthermore, it remains unclear why you ask European and domestic products together in your questions and do not differentiate here. Do you think that domestic products are assessed in the same way as European products? I strongly doubt this, as the literature on a wide variety of products repeatedly comes to the conclusion of how much national products are preferred by consumers.

Authors: Thank you for this observation. We agree! In fact, the question was structured in this way because the questionnaire was also designed to assess consumers' perception of the quality of European products, as it was created as part of a European project. Although it is always appropriate to separate concepts within a question (as we did for fruit and vegetables), a control question was nevertheless intended in the original questionnaire, which was mistakenly not considered during data processing. In the light of your observation, however, these answers have been incorporated into the results as a control question, as they indicate, for each country, the importance attached by consumers to the country of origin of the product they buy. In this regard, the results of these questions have been integrated into the revised manuscript. We added this information in Table 5, comparing the 4 clusters. In addition, we also commented on the difference between the three countries in terms of the respondents' first stated choice of the country of origin of fruit and vegetables during the purchasing process.

During the revision process, we realized that the percentages in the cells of Table 5 were wrong and revised all data and chi-square values with their respective significance levels.

Rev: STATISTICAL ANALYSIS:

Was an outlier analysis carried out before the actual cluster analysis? If not, why not? This should definitely be done!

It seems that no discriminant analysis was carried out to check the goodness of the identified clusters. This should definitely be done!

Authors:

We agree. Yes! We already performed the following analysis: first, outlier detection was performed using the box plot visualization tool; then, after cleaning the dataset, we performed a hierarchical CA to estimate the appropriate number of segments. This resulted in a scree plot, the node of which indicated that the ideal number of clusters would be 4. The 4-cluster segmentation was judged good in terms of cohesion and silhouette separation by employing the Two-steps method using both BIC and AIC indexes. Therefore, using the K-means clustering method, the 4 clusters were identified. Thank you for this comment: we have integrated the explanation into the text (see lines 200-206 of the revised manuscript).

Rev: RESULTS:

Table 2:

In order to assess the quality of the sample, it would be good if you could provide data on the distribution of the SD in the respective countries.

Authors: Thank you for this observation. We did not quite understand the request, sorry. Perhaps, it stems from the proposal of a cross-county analysis. However, by better clarifying the objective and removing any contradictory elements related to a cross-country analysis, Table 2 is already structured, we believe, to understand the composition of the sample as a whole and the differences by country in socio-demographic terms. We hope our interpretation of your request is correct.

Rev: Why did you choose a Varimax rotation? Please give reasons. Did you check the items regarding MSA and KMO? Were all items suitable for PCA?

Authors: thank you for this suggestion. The Varimax rotation was chosen because the variables were orthogonal belonging to different scales.

Below are the results of KMO and Bartlett's Test of Sphericity:

KMO and Bartlett's Test

Kaiser-Meyer-Olkin Measure of Sampling Adequacy.

,885

Bartlett's Test of Sphericity

Approx. Chi-Square

16549,681

df

120

Sig.

,000

As shown in the table, the suitability of the data for factor analysis was assessed by evaluating the correlations existing between the original variables by means of the Kaiser-Meyer-Olkin (KMO) statistics, which were found to be 0.885, exceeding the recommended value of 0.6 (Kaiser, 1974), and Bartlett's test of sphericity (Bartlett, 1954), which reached statistical significance. These values were added at the end of the PCA table. Thank you for the suggestion.

Rev: What about Cronbach's Alpha for the PCs? Are the found PCs reliable? Please check and add.

Authors: Yes, thank you very much for the statistical suggestions. We test the PC reliability and the results of the Cronbach's Alpha coefficients were added in the PCA table, together with the Pearson correlation coefficient for the PC4. Thank you!

Rev: Many items load on two or more PCs, which at first glance speaks against the results found with regard to the discriminatory power of the identified PCs.

Authors: Thank you for your observation. We checked and retested the variables by also performing the tests you suggested: we tested the validity of the PCA and the internal consistency of the individual PC. In fact, we restructured the table by selecting the correct items for each PC and checked and corrected the value of two loadings. The table was structured to make it clearer. We also re-evaluating the explanation and discussion about the PCA.

Rev: The rationale for naming the clusters is very brief. It would help readers who are less familiar with the method if it were explained in more detail how you arrived at the names. Please explicitly include Table 4 for this purpose.

Authors: thank you for this suggestion. We added the procedure explanation for clusters naming in the revised version of the manuscript.  

Rev: DISCUSSION

The discussion is well written for the present paper (consideration of the total sample). Here, after re-analysis, it would be important to write country-specifically.

The literature used here should also be considered, at least to a large extent, in the introduction.

Authors: Thank you very much for the suggestion. We have revised the discussion according to the edits included in the revised version of the manuscript, while maintaining the research framework of the original manuscript. Furthermore, as already suggested, some references and parts of the discussion have been included in the introductory section to introduce the research hypothesis.

Sincerely

Stefano Massaglia

Round 2

Reviewer 2 Report

Thank you very much for revising the paper so thoroughly. Both the revision and your additional letter to the reviewers did a very good job of showing how the paper has changed and why you decided to do certain things. I can well understand your explanation of why you have analysed the complete sample in this paper. Why don't you write this at the end of the paper under the heading "future analyses". Then it will also be understandable for the reader.
Good luck!

Author Response

Reviewer #2

Rev: Thank you very much for revising the paper so thoroughly. Both the revision and your additional letter to the reviewers did a very good job of showing how the paper has changed and why you decided to do certain things. I can well understand your explanation of why you have analysed the complete sample in this paper. Why don't you write this at the end of the paper under the heading "future analyses". Then it will also be understandable for the reader. Good luck!

Authors: We are very happy for your reply and grateful for the consideration. We have followed your suggestion by inserting a final paragraph in the text "future researches".

Thank you for your help during the manuscript revision process.

Sincerely

Stefano Massaglia